# Delivery of Oligonucleotides Using a Self-Degradable Lipid-Like Material

**DOI:** 10.3390/pharmaceutics13040544

**Published:** 2021-04-13

**Authors:** Hiroki Tanaka, Nae Takata, Yu Sakurai, Tokuyuki Yoshida, Takao Inoue, Shinya Tamagawa, Yuta Nakai, Kota Tange, Hiroki Yoshioka, Masatoshi Maeki, Manabu Tokeshi, Hidetaka Akita

**Affiliations:** 1Laboratory of DDS Design and Drug Disposition, Graduate School of Pharmaceutical Sciences, Chiba University, 1-8-1 Inohana, Chuo-ku, Chiba City, Chiba 260-0856, Japan; hiroki_tanaka8922@chiba-u.jp (H.T.); nae.lknossve.071221@gmail.com (N.T.); yu_sakurai@chiba-u.jp (Y.S.); 2Division of Molecular Target and Gene Therapy Products, National Institute of Health Sciences, 3-25-26 Tonomachi, Kawasaki-ku, Kawasaki, Kanagawa 210-9501, Japan; tokucox@nihs.go.jp (T.Y.); takao@nihs.go.jp (T.I.); 3DDS Research Laboratory, NOF CORPORATION, 3-3 Chidori-cho, Kawasaki-ku, Kawasaki City, Kanagawa 210-0865, Japan; shinya_tamagawa@nof.co.jp (S.T.); yuta_nakai@nof.co.jp (Y.N.); kota_tange@nof.co.jp (K.T.); hiroki_yoshioka@nof.co.jp (H.Y.); 4Division of Applied Chemistry, Faculty of Engineering, Hokkaido University, Kita 13, Nishi 8, Kita-ku, Sapporo, Hokkaido 060-8628, Japan; m.maeki@eng.hokudai.ac.jp (M.M.); tokeshi@eng.hokudai.ac.jp (M.T.)

**Keywords:** siRNA, antisense oligonucleotide, lipid nanoparticle

## Abstract

The world-first success of lipid nanoparticle (LNP)-based siRNA therapeutics (ONPATTRO^®^) promises to accelerate developments in siRNA therapeutics/gene therapy using LNP-type drug delivery systems (DDS). In this study, we explore the optimal composition of an LNP containing a self-degradable material (ssPalmO-Phe) for the delivery of oligonucleotides. siRNA or antisense oligonucleotides (ASO) were encapsulated in LNP with different lipid compositions. The hepatic knockdown efficiency of the target genes and liver toxicity were evaluated. The optimal compositions for the siRNA were different from those for ASO, and different from those for mRNA that were reported in a previous study. Extracellular stability, endosomal escape and cellular uptake appear to be the key processes for the successful delivery of mRNA, siRNA and ASO, respectively. Moreover, the compositions of the LNPs likely contribute to their toxicity. The lipid composition of the LNP needs to be optimized depending on the type of nucleic acids under consideration if the applications of LNPs are to be further expanded.

## 1. Introduction

Gene silencing using synthetic oligonucleotides such as a small-interfering RNA (siRNA) or an antisense oligonucleotide (ASO) is a promising approach for suppressing the expression of pathogenic proteins in cells/tissues. Since these oligonucleotides are easily degraded by enzymes and easily eliminated from the systemic circulation by renal clearance, nano-sized drug delivery systems (DDS) are useful for protection from degradation and the control of their pharmacokinetics [1]. In addition, the rational design of DDS could allow for more efficient endosomal escape. One of the most intensely investigated delivery systems is a lipid nanoparticle (LNP), which consists of artificial materials such as pH-sensitive cationic lipids or ionizable lipids [2]. In 2018, an LNP-based oligonucleotide therapeutic (ONPATTRO^®^) was approved as the first siRNA therapeutic by the Food and Drug Administration (FDA) and the European Medicines Agency (EMA) [3]. Based on this success, the use of an LNP-type DDS promises to further contribute to the development of siRNA therapeutics/gene therapy.

An important function of ionizable lipids is the pH-sensing ability of their tertiary amine groups. The neutral surface charge of the LNP at physiological pH is beneficial for avoiding undesired interactions with biomolecules. Once taken up by cells, the protonation of the tertiary amine groups in the acidic environment in endosomes results in the development of a cationic charge to the surface of the LNP. The cationic charge facilitates interactions between the LNP and the endosomal membrane, which promotes the subsequent escape of the oligonucleotides from the endosomal degradation pathway to the cytoplasm.

To achieve efficient gene regulation, oligonucleotides must be delivered to the cytoplasm. The release of oligonucleotides from the DDS is an important step in intracellular trafficking as well as in endosomal escape. To facilitate the release of the oligonucleotides, bio-degradable functional units are incorporated into the materials of the DDS [4]. The disulfide bond is frequently used in the design of bio-degradable materials in order to accelerate the release of oligonucleotides [5,6,7]. Since the intracellular concentration of glutathione (GSH, γ-Glu-Cys-Gly), a naturally produced reducing agent is, at most, 1000-fold higher intracellularly than in the extracellular environment, the disulfide bond can be selectively cleaved in the cytoplasm [8]. The rate of cleavage of a reduction-sensitive fluorescent probe that was directly microinjected into the cytoplasm indicated that the reaction reaches completion within approximately 2 min in the cell [9]. This spatial selectivity and high reactivity of the disulfide bond makes it suitable for use as a trigger that activates the destabilization of the DDS and the subsequent intracellular release of oligonucleotides.

For the delivery of oligonucleotides/genes, we previously developed an SS-cleavable and pH-activated lipid-like material (ssPalm) as a component of the LNP (LNP_ssPalm_) [10,11,12]. ssPalm contains two tertiary amine residues with fatty acid scaffolds that are bridged via a disulfide bond. These functional units are integrated into one molecule for facilitating both endosomal escape and the release of the loaded nucleic acids into the cytoplasm. We recently developed ssPalmO-Phe as a self-degradable derivative of ssPalm that is self-degraded in the intraparticle space by a specific hydrolytic reaction (Figure 1) [13]. The key functional units in its structure are phenyl esters that are located between the tertiary amines and fatty acid scaffolds. Since the phenyl esters are susceptible to the hydrolysis mediated by a thiol group, the intraparticle accumulation of thiol groups after the reduction of LNP_ssPalm_ triggers the subsequent hydrolysis of the phenyl esters. The hydrolysis results in the dissociation of the tertiary amines and hydrophobic scaffolds from the LNP. Of note, it was unexpectedly discovered that the insertion of an aromatic ring facilitates endosomal escape by enhancing membrane destabilizing activity. These features of LNP_ssPalmO-Phe_ function synergistically to promote the delivery of the loaded oligonucleotides/genes to the cytoplasm.

While ssPalmO-Phe was originally developed for the delivery of in vitro-transcribed messenger RNA (mRNA), we hypothesized that the self-degradation-driven release of the nucleic acids would also improve the efficiency of delivery of the other types of oligonucleotides. However, it is now recognized that the optimal lipid composition of the LNP is dependent on the type of nucleic acids being delivered [14,15]. In this study, we explored the optimal composition of LNP_ssPalmO-Phe_ for the hepatic delivery of siRNA by monitoring the membrane destabilizing ability of the system (LNP_ssPalmO-Phe_–siRNA), and the resulting LNP formulation was also then used for ASO delivery (LNP_ssPalmO-Phe_–ASO).

## 2. Materials and Methods

### 2.1. Animal Experiments

For the in vivo analysis of LNP_ssPalmO-Phe_–siRNA, C57BL6/J mice (male, 4–6 weeks of age) and ICR mice (male, 6–7 weeks of age) were purchased from Japan SLC, Inc (Shizuoka, Japan). The experimental protocols were reviewed and approved by the Chiba University Animal Care Committee in accordance with the Guide for Care and Use of Laboratory Animals. Ethical approval codes issued from the committee for this research include: 30–41. For the in vivo analysis of LNP_ssPalmO-Phe_–ASO, C57BL/6J mice (male, 5 weeks of age) were purchased from Charles River Laboratories Japan, Inc (Yokohama, Japan). The experimental procedures were reviewed and approved by the Institutional Animal Care and Use Committee of LSI Medience Corporation (Ibaraki, Japan). The ethical approval codes issued by the committee were 2020-0002 and 2020-0065.

### 2.2. Materials

Detailed information from suppliers is listed in the Supplementary Material Appendix A. The synthesis of ssPalmO-Ben (a non-degradable counterpart of ssPalmO-Phe) has been described in a previous manuscript [13]. ssPalmO-Phe (Product # COATSOME^®^ SS-OP), ssPalmO-P4C2 (Product # COATSOME^®^ SS-OC), ssPalmE-P4C2 (Product # COATSOME^®^ SS-EC), 1,2-dioleoyl-sn-glycero-3-phosphatidylcholine (DOPC; Product # COATSOME^®^ MC-8181), 1,2-distearyol-sn-glycero-3-phosphatidylcholine (DSPC; Product # COATSOME^®^ MC-8080), and 1-(monomethoxy polyethyleneglycol2000)2,3-dimyristoylglycerol (DMG-PEG2000; Product # SUNBRIGHT^®^ GM-020) were manufactured by NOF CORPORATION (Kanagawa, Japan). The suffix “P4C2” refers to 4-ethylpiperdine moieties that were incorporated as head groups into the structure. siRNA was purchased from Hokkaido System Science Co., Ltd. (Hokkaido, Japan). The sequence of the siRNA against factor VII (siFVII) has been reported previously [11]. A gapmer ASO targeting ApoB and one targeting Nr3c1 (also known as a glucocorticoid receptor) were purchased from Gene Design, Inc. (Osaka, Japan). The sequences of these ASOs, named ASOap and ASOgr, respectively, are provided in the Appendix A
Appendix A. Cholesterol and 6-(p-toluidino)-2-naphthalenesulfonic acid sodium salt (TNS) were purchased from Sigma-Aldrich (St. Louis, MO, USA). Amicon Ultra-4-100K centrifugal units and Amicon Ultra-15-100K centrifugal units were purchased form Merck Millipore (Darmstadt, Germany). Phosphate buffered saline (PBS) without Ca^2+^ and Mg^2+^ (PBS minus; PBS(−)) was purchased from NACALAI TESQUE, INC. (Kyoto, Japan). Ultrapure^TM^ Distilled water was purchased from Thermo Fisher Scientific (Waltham, MA, USA). Otsuka normal saline was purchased from Otsuka Pharmaceutical (Tokyo, Japan). All other reagents and chemicals were commercially available and were used without further purification.

### 2.3. Preparation of siRNA-Encapsulating LNP by a Microfluidic Device (siRNA)

For the preparation of the siRNA-encapsulating LNP used in “Effects of DOPC and cholesterol on hepatic knockdown efficiency” by an invasive lipid nanoparticle production (iLiNP) device [16], siFVII was diluted to a concentration of 7.5 μg/mL in 20 mM malic acid/NaOH buffer (600 μL, pH 3.0, with 30 mM NaCl). The lipid–ethanol solution was prepared at a concentration of 2.5 mM (400 μL). These solutions were mixed using the iLiNP device (total flow rate: 1 mL/min; flow ratio of water/ethanol: 3/2 (*v/v*)) by a syringe pump (Pump 33 Dual Drive System, Harvard Apparatus, Cambridge, MA, USA). The siRNA–lipid mixture (1 mL in total) was recovered and diluted with 3 mL of PBS(−). For the preparation of the siRNA-encapsulating LNP used in “Effects of the amount of PEG–lipid used” by NanoAssemblr^TM^ (Vancouver, BC, Canada), siFVII was diluted at a concentration of 0.69 μg/mL in 20 mM malic acid/NaOH buffer (650 μL, pH 3.0, with 30 mM NaCl). The lipid–ethanol solution was prepared at a concentration of 1.93 mM total lipids (350 μL). These solutions were mixed using a NanoAssemblr^TM^ (total flow rate: 20 mL/min; flow ratio of water/ethanol: 6.5/3.5 (*v/v*); total volume: 1 mL). The lipid–siRNA mixture (1 mL) was recovered and diluted with 3 mL of PBS(−). For the preparation of the siRNA-encapsulating LNP used in “Effects of the lipid/RNA ratio” and “Hemolytic activity in the presence of serum proteins” by a NanoAssemblr^TM^, siFVII was diluted at a concentration of 1.25 μg/mL in 20 mM malic acid/NaOH buffer (900 μL, pH 3.0, with 30 mM NaCl). The lipid–ethanol solution was prepared at a concentration of 4.5 mM total lipid (100 μL). These solutions were mixed using the NanoAssemblr^TM^ (total flow rate: 20 mL/min; flow ratio of water/ethanol: 9/1 (*v/v*); total volume: 1 mL). The siRNA–lipid mixture (1 mL) was recovered and diluted with 3 mL of PBS(−).

In all of these prepared samples, the external solution was replaced with PBS(−) by ultrafiltration using Amicon Ultra-4-100K (Darmstadt, Germany) centrifugal units (25 °C, 1000× *g*). The particle solution was diluted to an adequate concentration in PBS(−) before administration. Encapsulation efficiency and recovery of the siRNA were obtained by Quant-IT^TM^ RiboGreen^TM^ assay. The concentration of total siRNA and non-encapsulated siRNA were quantified after the LNP samples had been treated with Triton X-100 and without Triton X-100, as previously described [11]. The encapsulation efficiency was calculated as follows:Encapsulation efficiency (%) = 100 × ([total RNA]−[non-encapsulated RNA])/[total RNA]

### 2.4. Preparation of ASO-Encapsulating LNP by a Microfluidic Device

ASO was diluted to a concentration of 0.167 mg/mL in 20 mM malic acid/NaOH buffer (2400 μL, pH 3.0, with 30 mM NaCl). The lipid–ethanol solution was prepared at a concentration of 12.5 mM of total lipid (800 μL). These solutions were mixed using a NanoAssemblr^TM^ (total flow rate: 12 mL/min; flow ratio of water/ethanol: 3/1 (*v/v*)). The ASO–lipid mixture (3.2 mL in total) was recovered and diluted with 12.8 mL of PBS(−). The external solution was replaced by ultrafiltration with Ultrapure^TM^ distilled water using Amicon Ultra-15-100K centrifugal units (25 °C, 1000× *g*). The particle solution was diluted to an adequate concentration with Otsuka normal saline before administration. The recovery of the ASO was determined by a Quant-IT^TM^ OliGreen^TM^ assay. The OliGreen^TM^ assay was performed according to the manufacture’s protocol in the presence of 0.1% of Triton X-100. The calibration curve was generated using 125–1000 ng/mL ASO. Since the OliGreen^TM^ assay of the LNP–ASO in the absence of Triton X-100 did not produce reliable fluorescence, the encapsulation efficiency of the ASO was not calculated.

### 2.5. Evaluation of Surface pKa by TNS Assay

For the 6-(p-Toluidino)-2-naphthalenesulfonyl chloride (TNS) assay, 20 mM citric acid/NaOH buffer (with 150 mM NaCl, pH 3.0, 3.5, 4.0, 4.5, 5.0, 5.5), 20 mM sodium dihydrogen phosphate/NaOH buffer (with 150 mM NaCl, pH 6.0, 6.4, 6.8, 7.2, 7.6, 8.0), and 20 mM Tris/HCl buffer (with 150 mM NaCl pH8.5, 9.0, 9.5, 10.0) were prepared. TNS was dissolved at 0.6 mM in water as a stock solution. In the wells of a 96-well black plate, 2 μL of the TNS solution, 12 μL of the LNP solution (0.5 mM total lipid) and 186 μL of the each of the buffers above were mixed. After shaking the incubation mixture (400 rpm, 10 min), the fluorescence of the TNS (Ex: 321/Em: 447) was measured. The apparent pKa of the surface was calculated as the pH at which the LNP showed 50% of the maximum fluorescence.

### 2.6. Hemolysis Assay

pH-dependent hemolytic activity was used as an index of endosomal escape efficiency [17,18]. Phosphate buffered saline with malic acid (PMBS) buffer was prepared by dissolving DL-malic acid with PBS(−) to a concentration of 20 mM. The pH was then adjusted to pH 5.5, pH 6.5, and pH7.4 with NaOH solution. Whole blood from ICR mice was collected from the inferior vena cava in the presence of 0.5 μL of heparin sodium (5000 U/5 mL). Red blood cells were purified by washing the blood (1 mL) in 9 mL of PBS(−). The blood was centrifuged (4 °C, 400× *g*, 5 min) and the supernatant was discarded by aspiration. Washing was repeated 5 times to completely remove serum proteins. The red blood cells were then incubated with the LNP at pH 5.5, pH 6.5, and pH 7.4. The final concentration of the total lipid was from 1.56 μM to 400 μM. For hemolysis assays in the presence of serum proteins, the serum was collected and the protein concentration was quantified using the BCA Protein Assay kit (TAKARA Bio, Inc., Shiga, Japan) according to the manufacture’s protocol. The serum (10–60 μg of protein; final concentration: 20–120 μg/mL) was added to the red blood cells and the hemolysis assay was conducted.

### 2.7. FVII Assay

siFVII-encapsulating LNP_ssPalmO-Phe_ was diluted to the appropriate concentrations with PBS(−) and administered to C57BL6/J mice intravenously at the indicated dose. The volume was adjusted to 10 mL/kg. At 24 h after injection, 400 μL of blood was collected in the presence of 0.5 μL of heparin sodium (5000 U/5 mL) and stored on ice until used. The plasma concentration of FVII was determined using a colorimetric Biophen VII assay kit (Hyphen Biomed) according to the manufacturer’s protocol. The standard curve for Factor VII plasma levels was generated using plasma collected from non-treated mice. The baseline expression level of FVII was represented by the PBS-treated group.

### 2.8. Evaluation of Hepatotoxicity of LNP_ssPalmO-Phe_

LNP_ssPalmO-Phe_ was diluted to the appropriate concentrations with saline and intravenously administered to C57BL6/J mice at the indicated dose. The volume was adjusted to 10 mL/kg. At 96 h after injection, 500 μL of blood was collected and blood samples were processed to obtain serum, which was then stored below −80 °C until used. The level of serum aspartate aminotransferase (AST) and alanine aminotransferase (ALT) were measured using a biochemical autoanalyzer (DRI-CHEM 7000, Fujifilm Co., Tokyo, Japan, or TBA-2000FR, Canon Medical Systems Co., Tochigi, Japan).

### 2.9. Evaluation of the Knockdown Efficiency of ASOs

ASOap, ASOgr, ASOap-encapsulating LNP_ssPalmO-Phe_ and ASOgr-encapsulating LNP_ssPalmO-Phe_ were diluted to the appropriate concentrations with saline and administered to C57BL6/J mice intravenously at the indicated dose. The volume was adjusted to 10 mL/kg. At 96 h after injection, livers were isolated from mice and stored in RNAlater RNA Stabilization Reagent (Qiagen, Valencia, CA, USA) overnight. Livers were homogenized using a BioMasher II (Nippi, Tokyo, Japan). Total RNA was isolated from mouse liver tissues using the RNeasy Mini Kit (Qiagen, Valencia, CA, USA) according to the manufacturer’s protocol. qRT-PCR was performed using a One Step TB Green PrimeScript PLUS RT-PCR Kit (TAKARA Bio, Inc., Shiga, Japan) and analyzed with a 7500 Fast Real-Time PCR System (Applied Biosystems, Foster City, CA, USA). The primers used in this study are shown in the Appendix A. The level of target gene expression was normalized to that of mouse Glyceraldehyde 3-phosphate dehydrogenase (GAPDH). Hepatotoxicity evaluation was performed as described above.

## 3. Results

### 3.1. Optimization of the Lipid Composition

#### 3.1.1. Effects of the Phospholipid and Cholesterol

The structure of the lipid components in LNP_ssPalmO-Phe_ are summarized in Figure 1. In addition to ssPalmO-Phe as an ionizable lipid, DOPC, cholesterol, and PEG–lipid were incorporated to increase the stability of LNP_ssPalmO-Phe_. The PEG–lipid is incorporated to improve the colloidal stability of LNP_ssPalmO-Phe_ by forming a hydrophilic layer on the surface of the particles. In a previous report, it was revealed that the optimal composition for LNP_ssPalmO-Phe_ was ssPalmO-Phe/DOPC/Chol = 52.5/7.5/40 with an additional 3% PEG–lipid. For siRNA delivery by another derivative of ssPalm (ssPalmE-P4C2, as described in Section 3.2.1), however, the optimal composition for the hepatic delivery of the siRNA was ssPalm/Chol = 70/30 [11]. Based on these previously published results, the knockdown efficiency against a liver specific gene (FVII) was evaluated using the LNPs in which 30–60% of cholesterol had been incorporated. In addition, the 7.5% of ssPalmO-Phe was replaced with DOPC in the 30% and 40% cholesterol groups (Figure 2). The serum concentration of the FVII at a dose of 0.02 mg/kg siFVII was evaluated 24 h after administration. As a result, LNP_ssPalmO-Phe_ formed with less cholesterol content exhibited a higher knockdown efficiency. The most efficient lipid composition was determined to be ssPalmO-Phe/Cholesterol = 70/30 without DOPC.

#### 3.1.2. Effects of the PEG–Lipid and Lipid/siRNA Ratio

Although the PEG–lipid is important in terms of conferring colloidal stability to the LNP_ssPalmO-Phe_ construct, modifying the surface of the LNP with such a hydrophilic polymer is accompanied by the risk of decreasing the efficiency of the delivery system [19,20]. Since the original composition of LNP_ssPalmO-Phe_ contained 3% PEG–lipid, the amount of PEG–lipid was optimized based on the knockdown efficiency for the preparation. Particle data are summarized in Table 1. Reducing the PEG–lipid content from 3.0% to 1.0% resulted in an improvement in the FVII knockdown activity (Figure 3a). On the other hand, modification with 0.5% and 1.0% PEG–lipid resulted in increased particle size. This observation indicates that LNP_ssPalmO-Phe_ with less than 1.0% PEG–lipid would be unstable. It is plausible that the minimum amount of the PEG–lipid needed to produce a stable particle preparation was 1.5% total lipids. Increasing the lipid/siRNA ratio (nmol/μg) further improved the knockdown efficiency (Figure 3b). In conclusion, the most efficient formulation of LNP_ssPalmO-Phe_ for siRNA delivery was determined to be ssPalmO-Phe/Chol/DMG-PEG2000 = 70/30/1.5 at a lipid/siRNA ratio of 400 nmol/μg.

#### 3.1.3. Evaluation of the ED_50_ Value

Knockdown efficiency of the optimized LNP_ssPalmO-Phe_ was evaluated with the 50% effective dose (ED_50_) as a reference. The dose–response curve showed that the ED_50_ of the hepatic knockdown by LNP_ssPalmO-Phe_ was 4.4 μg/kg siRNA (0.0044 mg/kg, Figure 4). At a dose of 40 μg/kg siRNA, the knockdown efficiency was more than 95% (98.7 ± 1.3%).

### 3.2. Effect of the Lipid Structure on the Knockdown Efficiency and Endosomal Escape

#### 3.2.1. Effect of the Linker and Hydrophobic Scaffold

The above data indicated that LNP_ssPalmO-Phe_ showed a potent delivery efficiency of the siRNA to the liver. Meanwhile, it was previously demonstrated that the hydrophobic scaffolds of the ionizable lipids have a profound effect on delivery efficiency [11]. To further analyze the function of the phenyl linker and the oleic acid scaffold, the knockdown efficiency of ssPalms with different hydrophobic scaffolds was compared (Table 2 and Figure 5a). The ssPalm used in this comparison included an ssPalm with oleic acid scaffolds without a linker (ssPalmO-P4C2), an ssPalm with oleic acid and a non-degradable aromatic ring-type linker (benzyl ester, ssPalmO-Ben) and an ssPalm with a vitamin E scaffold (ssPalmE-P4C2) that was previously reported to have the ability to deliver siRNA to the liver. LNP_ssPalmE-P4C2_ and LNP_ssPalmO-P4C2_ showed a comparable knockdown efficiency (15.4 ± 8.7% and 19.2 ± 4.4%, respectively). In contrast, the knockdown efficiency of LNP_ssPalmO-Ben_ (68.3 ± 2.0%) was significantly improved compared to LNP_ssPalmO-P4C2_. The knockdown efficiency of LNP_ssPalmO-Phe_ (81.3 ± 0.5%) was further improved compared to that of LNP_ssPalmO-Ben_ (Figure 5b).

#### 3.2.2. Evaluation of Hemolytic Activity

Although ssPalmO-Phe, ssPalmO-Ben, and ssPalmO-P4C2 all contain oleic acid scaffolds, these materials showed significantly different knockdown efficiencies due to their linker structures. To analyze the effects of the linkers, the hemolytic activity of these materials was compared. The hemolytic activity in the physiological environment was originally used as an index of toxicity caused by intravascular hemolytic events. On the other hand, the hemolytic activity of the LNPs in an acidic environment can be accepted as an index of endosomal escape efficiency since that value reflects the membrane-destabilizing ability of the DDS against the endosomal membrane [17,18]. The hemolytic activity of LNP_ssPalmO-Phe_ and LNP_ssPalmO-Ben_, both of which contain aromatic ring moieties, was higher than that of LNP_ssPalmO-P4C2_ (Figure 6a). The surface charge of LNP_ssPalm_ was evaluated by a fluorescent probe 6-(p-toluidino)-2-naphthalenesulfonic acid sodium salt (TNS)-based assay. TNS fluoresces when it is inserted into a hydrophobic environment. In the case of an LNP that contains ionizable lipids, the protonation of the ionizable lipids at an acidic pH can be monitored by the increase in fluorescence of TNS. The apparent surface pKa of LNP_ssPalm_ was calculated at the pH at which the TNS showed 50% of maximum fluorescence. The pKa value of the LNP was 5.97 ± 0.06 for LNP_ssPalmO-Phe_, 5.92 ± 0.04 for LNP_ssPalmO-ben_, and 5.88 ± 0.02 for LNP_ssPalmO-P4C2_, respectively (Figure 6b). Since no large differences were found for the extent of protonation of the surface, the improvements in hemolytic activity of LNP_ssPalmO-Phe_ and LNP_ssPalmO-Ben_ can be attributed to the presence of the aromatic ring. The difference between ssPalmO-Phe and ssPalmO-Ben cannot be explained by the endosomal escape efficiency. The slightly improved hepatic knockdown efficiency of LNP_ssPalmO-Phe_ reflects the enhanced release of siRNA, which was accelerated by the self-degradability of the phenyl linker.

#### 3.2.3. Hemolytic Activity in the Presence of Serum Proteins

Once injected into the systemic circulation, serum proteins deposit on the surface of the LNP [21,22,23]. Thus, the hemolytic activity in the presence of serum proteins would more accurately reflect the in vivo situation. Hemolytic activity was evaluated in the presence of 20–120 μg/mL of serum proteins. As a control, DLin-MC3-DMA (MC3), an ionizable lipid used in ONPATTRO^®^, was used as a component of the LNP. To evaluate the contribution of ionizable lipids, the composition of the LNP was adjusted to that of the LNP containing MC3 (ionizable lipids/distearoyl-sn-glycero-phosphatidyl-choline (DSPC)/Chol/DMG-PEG2000 = 50/10/38.5/1.5) [24,25]. The dose–response curve for hemolytic activity was evaluated by changing the total lipid concentration. The presence of serum proteins tended to decrease the hemolytic activity by inhibiting the interaction between the surface of the LNP and red blood cells (Figure 7). It was unexpectedly revealed that LNP_ssPalmO-Phe_ had a higher resistance against the presence of serum proteins than LNP_MC3_. When we compared hemolytic activity at 100 μM lipid concentration (the same condition as Figure 5), the presence of 30 μg and 60 μg of serum proteins resulted in a decrease in the hemolytic activity of LNP_MC3_ from 84.9 ± 9.4% to 10.3 ± 2.8% and −0.3 ± 1.1%, respectively. On the other hand, for LNP_ssPalmO-Phe_, the hemolytic activity was slightly decreased from 69.5 ± 8.2% to 52.0 ± 5.3% and 30.1 ± 5.9%, respectively. In this analysis, the concentration of serum proteins ranged from 10 μg/mL to 120 μg/mL. The relative hemolytic activity at each protein/lipid ratio was plotted (Appendix A). The relative hemolytic activity of MC3 decreased at smaller protein/lipid ratios compared to ssPalmO-Phe, which is consistent with our conclusion.

### 3.3. Application for the Delivery of ASO

#### 3.3.1. Composition Modification for ASO Delivery

LNP_ssPalmO-Phe_ was applied for the delivery of ASO. To this end, we used an locked nucleic acid (LNA) gapmer ASO targeting *ApoB* (ASOap) and one targeting *Nr3c1* (also known as the glucocorticoid receptor) (ASOgr) as ASO therapeutic models [26,27]. We initially hypothesized that the composition of ssPalmO-Phe/Chol = 70/30, the optimal composition for siRNA delivery, could be applied to ASO. However, it was revealed that LNP_ssPalmO-Phe_ containing ASO is highly unstable and susceptible to aggregation. To stabilize LNP_ssPalmO-Phe_ containing ASO, the lipid composition and lipid/ASO ratio were modified. It was ultimately found that the composition of ssPalmO-Phe/DOPC/Chol = 40/20/40 with an additional 1.5% of DMG-PEG2000 and a lipid/ASO ratio of 25 nmol/μg was suitable for preparing stable LNP_ssPalmO-Phe_ containing ASO (Table 3). The knockdown efficiency of LNP_ssPalmO-Phe_–ASOap and LNP_ssPalmO-Phe_–ASOgr was evaluated in the mouse liver at doses in the range of 0.005–0.37 mg/kg. As a comparison, naked ASO (ASOap or ASOgr) was injected in the range of 0.37–30 mg/kg as a control. The dose–response curve showed that the 50% inhibitory concentration (IC_50_) for hepatic knockdown by ASOap and LNP_ssPalmO-Phe_–ASOap was 1.41 mg/kg and 0.0097 mg/kg, respectively (Figure 8a), indicating that the knockdown efficiency of LNP_ssPalmO-Phe_–ASOap was 146 times higher than ASOap. For ASOgr, the IC_50_ for hepatic knockdown by ASOgr and LNP_ssPalmO-Phe_–ASOgr was 3.61 mg/kg and 0.092 mg/kg, respectively (Figure 8b), showing that the knockdown efficiency of LNP_ssPalmO-Phe_–ASOgr was 39.3 times higher than ASOgr. These observations indicate that LNP_ssPalmO-Phe_ significantly improved the delivery efficiency of ASO to the liver.

#### 3.3.2. Comparison of Hepatotoxicity between ASO and LNP_ssPalmO-Phe_–ASO

ASOap and ASOgr are gapmer ASOs that have been reported to induce hepatotoxicity in mice [26,27]. Thus, we simultaneously analyzed the changes in the degree of hepatotoxicity of ASOap and ASOgr when they were encapsulated in LNP_ssPalmO-Phe_. LNP_ssPalmO-Phe_–ASOap induced severe hepatotoxicity when injected at a dose of 0.37 mg/kg, while no changes in serum AST and ALT levels for ASOgr at the same dose were found compared to the saline group (Figure 9a). Similar results were observed in the comparison between LNP_ssPalmO-Phe_–ASOgr and ASOgr (Figure 9b). These data support the finding that LNP_ssPalmO-Phe_ significantly improved the delivery efficiency of ASO to the liver, which enables the hepatotoxicity of ASOs to be detected at a lower dosage.

### 3.4. Safety of Empty LNP_ssPalmO-Phe_

To confirm that the hepatotoxicity induced by LNP_ssPalmO-Phe_–ASO was caused by the ASO and not from the LNP-forming component, we evaluated the safety of ASO-free LNP_ssPalmO-Phe_. Mice were administered empty LNP_ssPalmO-Phe_ and hepatotoxicity was then assessed. Mice were intravenously injected with LNP_ssPalmO-Phe_ at doses in the range of 49–782 mg/kg of total lipids (corresponding to 29–470 mg/kg ssPalmO-Phe), which theoretically can contain 2.5–40 mg/kg ASO. There was no significant change in AST and ALT levels at any concentration compared to the saline group (Figure 10). These data indicate that LNP_ssPalmO-Phe_ did not induce hepatotoxicity in the concentration range used for the application of ASO.

## 4. Discussion

In this study, the optimal composition of LNP_ssPalmO-Phe_ was explored in an attempt to achieve more efficient delivery of a series of oligonucleotides. The findings revealed that the optimal composition for siRNA delivery was ssPalmO-Phe/Chol = 70/30, and that for the ASO delivery, the optimal composition was ssPalmO-Phe/DOPC/Chol = 40/20/40, respectively. Both compositions contained an additional 1.5% of DMG-PEG2000. These compositions were different from the composition (ssPalmO-Phe/DOPC/Chol = 52.5/7.5/40) that was optimized for the delivery of mRNA [13]. This observation strongly supports the current understanding that the composition of the LNP needs to be optimized for each therapeutic modality being considered [14,15]. The differences can be partially attributed to the chemical properties of each nucleic acid. The mRNA molecule is the most vulnerable molecule among these nucleic acids, since mRNA basically does not contain chemical modification in its phosphodiester backbone and ribose structure. The mRNA molecule can easily be degraded by nucleases in the extracellular environment. Hydrolysis of the backbone would result in complete loss of its function and would cause its subsequent degradation by nucleases. Thus, the incorporation of 7.5% DOPC and 40% cholesterol seems to be a prerequisite for protecting it by improving membrane integrity.

In the case of siRNA, the molecule is chemically more stable than mRNA, since its ribose structure has been artificially modified. However, for eliciting the maximum potency of the siRNA, it is important that the siRNA reaches the cytoplasm in double-stranded form [28]. It is known that the optimal pKa of LNP systems for the hepatic delivery of siRNA was 6.2–6.5 [24]. These observations suggest that the siRNA may need to be released in an intact form from the endosomal compartments at an early stage of the endocytosis process. The results shown in Figure 5a suggest that the optimal composition of LNP_ssPalmO-Phe_ for the delivery of siRNA has a suitable membrane-destabilizing ability, since hemolytic activity was observed at pH 6.5. To achieve a high endosomal escape efficiency, the optimal composition for siRNA required a large amount of ionizable lipids (up to 70% of the total lipids).

As LNP_ssPalmO-Phe_ containing ASO is unstable, the ratio of DOPC needed to be increased to 20% of the total lipids. Although ASOap (13 mer) and ASOgr (14 mer) have a similar molecular size and charge, the Zeta-potential and ASO recovery ratio for LNP_ssPalmO-Phe_–ASOap and LNP_ssPalmO-Phe_–ASOgr were different (Table 3). These differences can be partly explained by the structure of ASO. While siRNA and mRNA molecules have a rigid double-stranded structure as a result of inter- and intra-molecular base pairing, ASO has a flexible single-stranded structure. This indicates that the nucleobases of ASO, in addition to its phosphorothioate backbone, can interact with ssPalm. In other words, in the case of single-stranded ASO, the sequence and/or nucleobase content would have a profound effect on the properties of the LNP. This sequence dependency should be taken into account in the case of ASO delivery using DDS. Since the ASO itself was highly stable and relatively hydrophobic because of its chemically modified structure, it is expected that the ASO would reach the cytoplasm more readily once taken up by the cells. It is well known that a neutral LNP is taken up by hepatocytes via apolipoprotein E (ApoE) and low-density lipoprotein receptor (LDLR)-dependent endocytosis [29]. This hepatic uptake mediated by these endogenous proteins most likely plays a key role in the hepatic delivery of ASO. To ensure efficient extravasation through the fenestra and efficient uptake by the liver cells, the use of a lipid composition that permits stable and small particles to be prepared is important for the delivery of ASO. Collectively, the lipid composition of the LNP should be determined by taking the critical steps for each nucleic acid into consideration: extracellular stability of the mRNA, endosomal escape of the siRNA, and cellular uptake of the ASO, respectively.

The findings shown in Figure 5a revealed that the incorporation of an aromatic ring improved endosomal escape efficiency. It was unexpectedly revealed that LNP_ssPalmO-Phe_ conferred resistance against the effect of serum proteins compared to the conventional ionizable lipids (Figure 7 and Appendix A). Since this resistance against serum proteins was also observed for ssPalmO-P4C2 (Appendix A), we concluded that this property was conferred by the structural characteristics of ssPalm and not by the insertion of a phenyl ester. As the bond angle of the disulfide bond is located at the center of the chemical structure, the conformation of ssPalm was calculated as a linear form compared to that of conventional lipids with head–tail structures. The distribution of the angles formed by the two oleic acid scaffolds ranges from 30° to 150°, and for ssPalmO-P4C2, the most frequent angle was 140° (open conformation) [13]. Thus, the surface of LNP_ssPalm_ might be hydrophobic compared to an LNP that is composed of conventional lipids. This difference in the surface properties of LNP_ssPalm_ would be expected to result in changes in molecular interactions and the quantity of the proteins on the surface of the particles. However, the detailed mechanisms of the resistance to serum proteins remains to be clarified.

The findings reported herein reveal that the encapsulation of ASO in LNP_ssPalmO-Phe_ significantly improved the knockdown efficiency of the preparation (Figure 8). In parallel, the liver toxicity of the ASO was also observed at lower doses (Figure 9). The dose for the total lipids for LNP_ssPalmO-Phe_–ASOap and LNP_ssPalmO-Phe_–ASOgr was 0.41–30 mg/kg and 0.14–10.1 mg/kg, respectively. Since there was no evidence of toxicity for the empty particle in the concentration range of 49–782 mg/kg total lipids (29–470 mg/kg ssPalmO-Phe), the toxicity observed for LNP_ssPalmO-Phe_-ASO was due to the improved efficiency of the delivery of these toxic ASOs (Figure 10). On the other hand, we previously reported that the administration of LNP_ssPalmO-Phe_ with an optimal composition for mRNA at a dose of more than 350 mg/kg total lipids (>260 mg/kg ssPalmO-Phe) induced an elevation in serum AST/ALT levels [13]. These observations indicate that the liver toxicity of LNPs_sPalmO-Phe_ was not simply determined by the dose of the ssPalmO-Phe molecule, but also reflects the difference in the lipid composition. The ratio of ssPalmO-Phe/DOPC was different between the lipid compositions for ASO (ssPalmO-Phe/DOPC/Chol = 40/20/40) and mRNA (ssPalmO-Phe/DOPC/Chol = 52.5/7.5/40). It is generally thought that phospholipids in the LNP are aligned on the surface of the nanoparticle [25]. Due to its amphiphilicity and overall cylindrical shape, these phosphatidylcholines confer stability to the surface of LNP_ssPalmO-Phe_. Although the stability of a nanoparticle is important in maintaining its dispersibility, excess stabilization could result in poor transfection activity. On the other hand, a strong membrane destabilizing ability is also accompanied by the risk of adverse responses due to the leakage of the endosomal compartment into the cytoplasm. The destabilization of the endosome, also known as endosomolysis, results in cell death via apoptosis/necrosis depending on the severity of the membrane destabilization [30,31,32]. This toxic event can be induced by exogenous molecules such as a crotamine derivatives and can activate a cathepsin–caspase axis [33,34].

Since the optimal composition of an LNP depends on the physicochemical/biological properties of each nucleic acid, it is difficult to present a general principle for the process of optimization for different types of nucleic acids. In the case of RNAs (siRNA and mRNA), endosomal escape is still considered to be a major barrier that inhibits cytoplasmic delivery. Thus, hemolytic activity, the index of endosomal escape efficiency, needs to be prioritized. Since the phospholipid content is difficult to determine using in vitro experiments, experimental verification in vivo would be needed to develop an optimal LNP. However, hepatic damage may also depend on the membrane destabilizing activity of the LNP. Thus, a parallel evaluation of the hemolytic activity and the hepatic toxicity is needed to confirm the efficacy and safety. On the other hand, ASO is a unique material because of its innate membrane permeability characteristics. In this case, stability in the circulation should be a more important parameter to be optimized for hepatic targeting, as avoiding renal clearance is the main purpose of the use of an LNP. Thus, the optimal composition depends on the combination of the ASO being used and its target.

## 5. Conclusions

In this manuscript, we report on the optimal lipid compositions of LNPs for the delivery of siRNA and ASO. The difference in the optimal lipid compositions can be attributed to the intracellular trafficking/pharmacokinetics of the nucleic acids. Since the membrane destabilizing ability of ssPalmO-Phe is beneficial for overcoming the plasma/endosomal membrane, LNP_ssPalmO-Phe_ can be used to deliver both nucleic acids. The lipid composition should be carefully selected from the point of view of both delivery efficiency and safety.

## 6. Patents

H. Tanaka, N. Takata, Y. Nakai, K. Tange, H. Yoshioka, and H. Akita are the inventors of the patent pending (WO2013/073480 and WO2016/121942) on the ssPalm chemicals.

## Figures and Tables

**Figure 1 pharmaceutics-13-00544-f001:**
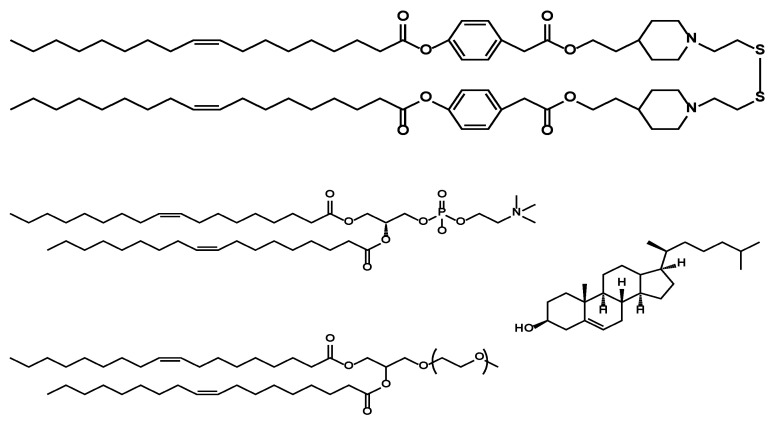
Chemical structures of the lipid components of lipid nanoparticle (LNP). The LNP containing ssPalmO-Phe was referred as LNP_ssPalmO-Phe_. LNP_ssPalmO-Phe_ contained ssPalmO-Phe, dioleoyl-sn-glycero-phosphatidylcholne (DOPC), cholesterol, and polyethylene glycol (PEG)-conjugated lipid.

**Figure 2 pharmaceutics-13-00544-f002:**
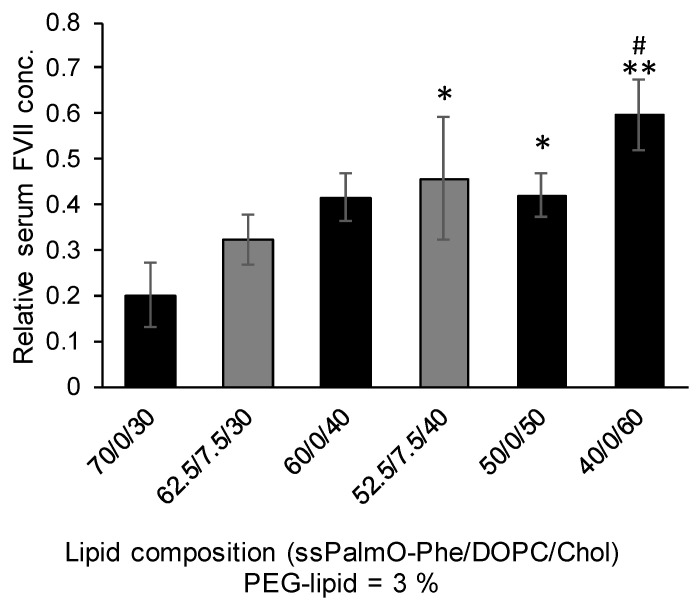
Effects of DOPC and cholesterol on hepatic knockdown efficiency (3 mice/group). The siRNA against factor VII (siFVII) was encapsulated in LNP_ssPalmO-Phe_ at the indicated lipid composition. All of the LNP_ssPalmO-Phe_ constructs contained an additional 3% of 1-(monomethoxy polyethyleneglycol2000)2,3-dimyristoylglycerol (DMG-PEG2000). Mice were injected with LNP_ssPalmO-Phe_ in the tail vein; the dose was 0.02 mg/kg siRNA. The serum concentration of FVII was quantified 24 h after injection. The relative serum concentration was normalized to the PBS group. Statistical analysis was performed by one-way ANOVA followed by Tukey–Kramer test; *: *p* < 0.05 against 70/0/30, **: *p* < 0.01 against 70/0/30, ^#^: *p* < 0.05 against 62.5/7.5/30.

**Figure 3 pharmaceutics-13-00544-f003:**
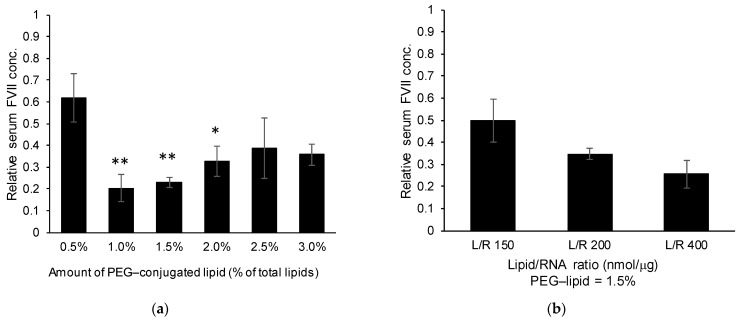
(**a**) Effects of the amount of PEG–lipid used (3 mice/group). siFVII was encapsulated in LNP_ssPalmO-Phe_. The composition of the lipid was ssPalmO-Phe/cholesterol = 70/30 with the indicated amount of PEG–lipid. Statistical analyses were performed by one-way ANOVA followed by Tukey–Kramer test; *: *p* < 0.05 against 0.5% PEG modification, **: *p* < 0.01 against 0.5% PEG modification. (**b**) Effects of the lipid/RNA ratio (3 mice/group). LNP_ssPalmO-Phe_ containing ssPalmO-Phe/cholesterol = 70/30 with 1.5% DMG-PEG2000 was prepared with different ratios of lipid/siRNA (nmol/μg). Mice were intravenously injected with LNP_ssPalmO-Phe_ via the tail vein. The doses were 0.02 mg/kg siRNA and 0.01 mg/kg siRNA for (**a**) and (**b**), respectively. At 24 h after injection, the serum concentration of FVII was quantified. The relative serum concentration was normalized to the PBS group.

**Figure 4 pharmaceutics-13-00544-f004:**
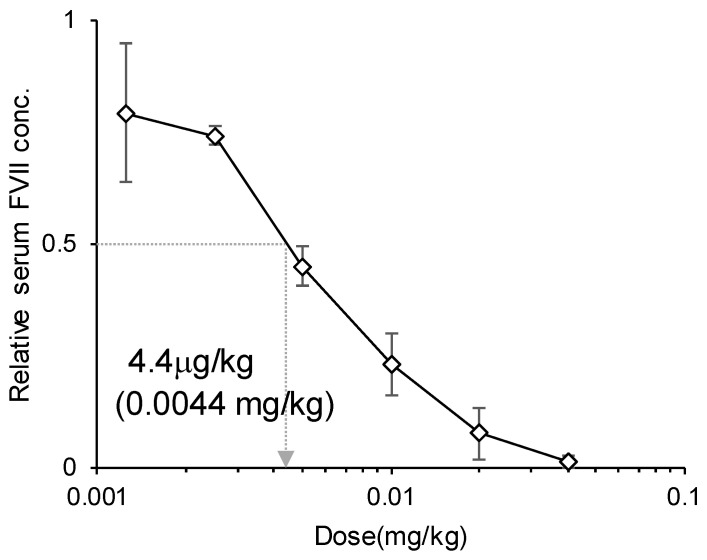
Dose–response curve of the knockdown efficiency (3 mice/group). Mice were intravenously injected with LNP_ssPalmO-Phe_ containing ssPalmO-Phe/cholesterol = 70/30 with 1.5% DMG-PEG2000, which was prepared with a lipid/RNA ratio of 400 nmol/μg, via the tail vein. The doses were 1.25–40 μg/kg siRNA. The 50% effective dose (ED_50_) value of LNP_ssPalmO-Phe_ was calculated as the dose at which LNP_ssPalmO-Phe_ showed 50% knockdown of FVII.

**Figure 5 pharmaceutics-13-00544-f005:**
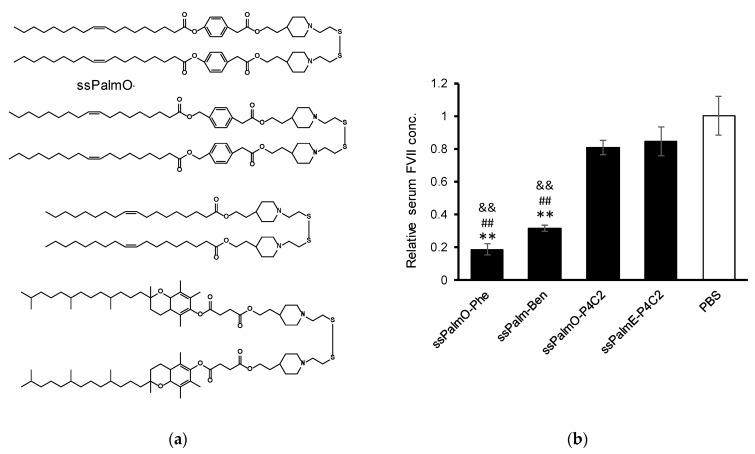
(**a**) Chemical structures of ssPalm used in comparing the effectiveness of hydrophobic scaffolds. ssPalmO-P4C2 contained piperidine head groups and oleic acid scaffolds without linkers. ssPalmE-P4C2 contained vitamin E (α-tocopherol) as a hydrophobic scaffold instead of oleic acid [11]. (**b**) Effects of the hydrophobic scaffolds on knockdown efficiency (3 mice/group). LNP_ssPalm_ was formed with the indicated ssPalm. The lipid composition was ssPalm/Chol = 70/30 with an additional 3% of DMG-PEG2000. The dose was 0.02 mg/kg siRNA. Statistical analyses were performed by one-way ANOVA followed by Tukey–Kramer test; **: *p* < 0.01 against PBS, ^##^: *p* < 0.01 against ssPalmO-P4C2, ^&&^: *p* < 0.01 against ssPalmE-P4C2.

**Figure 6 pharmaceutics-13-00544-f006:**
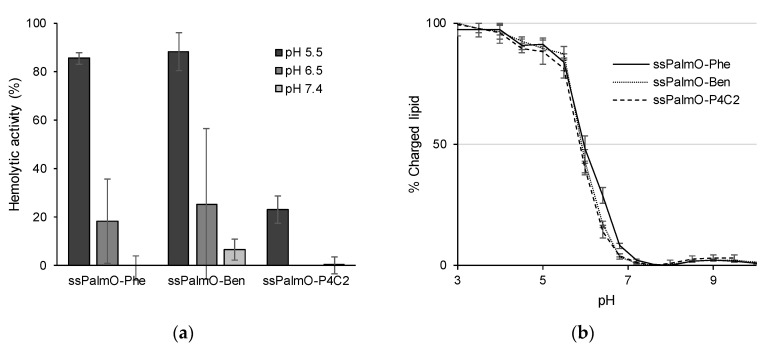
(**a**) Hemolytic activity of LNP_ssPalm_ containing oleic acid scaffolds. Hemolytic activity at pH 5.5, pH 6.5, and pH 7.4 was evaluated. Total lipid concentration was 100 μM. The hemolytic activity is shown as a percentage of the positive control (Triton X-100). (**b**) Surface charge of LNP_ssPalm_ containing oleic acid scaffolds. The ratio of charged lipids was evaluated by 6-(p-Toluidino)-2-naphthalenesulfonyl chloride (TNS) assay. The apparent pKa value of LNP_ssPalm_ was calculated at the pH at which the surface of LNP_ssPalm_ showed 50% of the maximum fluorescence.

**Figure 7 pharmaceutics-13-00544-f007:**
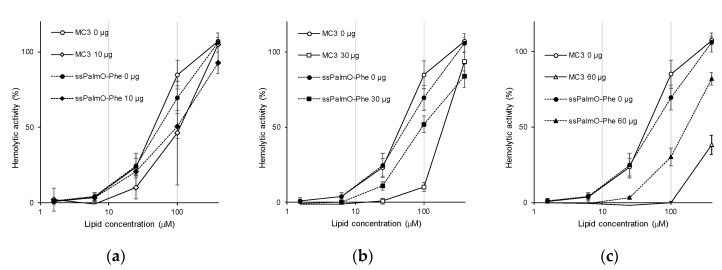
Hemolytic activity in the presence of serum proteins. Hemolytic activity at pH 5.5 was evaluated using LNP_ssPalmO-Phe_ (filled symbols) or LNP containing DLin-MC3-DMA (LNP_MC3_ (open symbols). The lipid composition was adjusted to the previous literature values (ionizable lipids/DSPC/Chol/PEG = 50/10/38.5/1.5) [24,25]. Hemolytic activity was assessed in the presence of (**a**) 10 μg, (**b**) 30 μg and (**c**) 60 μg of serum proteins, respectively. Hemolytic activity in the absence of serum proteins as a control is shown in all charts.

**Figure 8 pharmaceutics-13-00544-f008:**
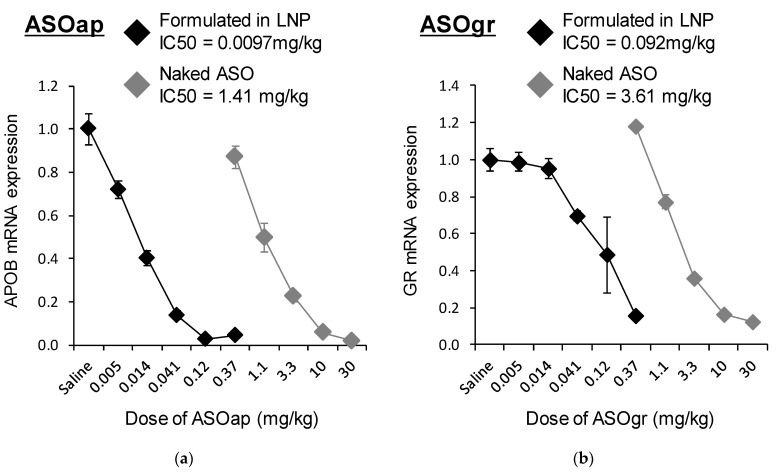
Dose–response curve for the knockdown efficiency of ASO (4 mice/group). LNP_ssPalmO-Phe_ (ssPalmO-Phe/DOPC/Chol = 40/20/40 with an additional 1.5% DMG-PEG2000) was prepared with a lipid/ASO ratio of 25 nmol/μg. Mice were injected with each of the LNP_ssPalmO-Phe_–ASOap and LNP_ssPalmO-Phe_–ASOgr preparations at doses in the range of 0.005–0.37 mg/kg, while ASOap and ASOgr were injected at doses in the range of 0.37–30 mg/kg. The 50% inhibitory concentration (IC_50_) value of LNP_ssPalmO-Phe_–ASOap, LNP_ssPalmO-Phe_–ASOgr, ASOap and ASOgr were calculated as the dose at which they showed 50% knockdown of each target gene. LNP_ssPalmO-Phe_–ASOap and ASOap: ASO targeting *ApoB* (**a**); LNP_ssPalmO-Phe_–ASOgr and ASOgr: ASO targeting *Nr3c1* (**b**). The relative mRNA expression was normalized to the saline group.

**Figure 9 pharmaceutics-13-00544-f009:**
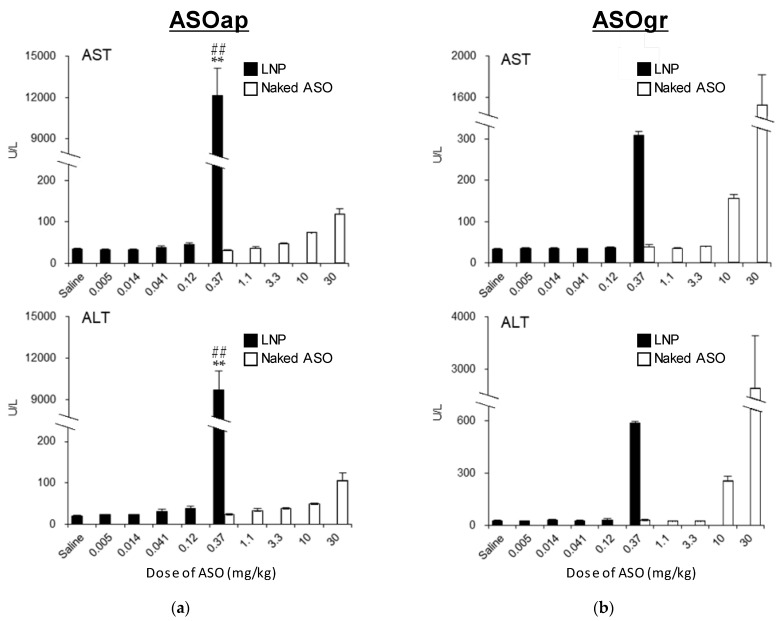
Hepatotoxicy of LNP_ssPalmO-Phe_–ASOap and LNP_ssPalmO-Phe_–ASOgr (4 mice/group). LNP_ssPalmO-Phe_ (ssPalmO-Phe/DOPC/Chol = 40/20/40 with an additional 1.5% of DMG-PEG2000) was prepared with a lipid/ASO ratio of 25 nmol/μg. Mice were injected with LNP_ssPalmO-Phe_–ASOap and LNP_ssPalmO-Phe_–ASOgr at doses in the range of 0.005–0.37 mg/kg, while ASOap and ASOgr were injected at doses in the range of 0.37–30 mg/kg. Serum aspartate aminotransferase (AST) and alanine aminotransferase (ALT) levels were measured 96 h after administration. LNP_ssPalmO-Phe_–ASOap and ASOap (**a**), LNP_ssPalmO-Phe_–ASOgr and ASOgr (**b**). Statistical analysis was performed by one-way ANOVA followed by Tukey–Kramer test; **: *p* < 0.01 against saline, ##; *p* < 0.01 against 0.37 mg/kg ASOap.

**Figure 10 pharmaceutics-13-00544-f010:**
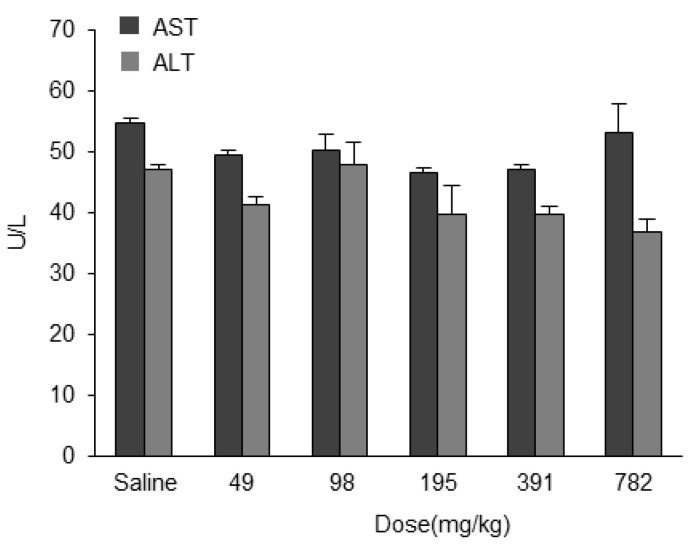
Safety of LNP_ssPalmO-Phe_. LNP_ssPalmO-Phe_ without ASO was intravenously injected into the the tail vein of mice at doses ranging from 49–782 mg/kg of total lipids (corresponding to 29–470 mg/kg ssPalmO-Phe). Serum AST and ALT levels were measured 96 h after administration.

**Table 1 pharmaceutics-13-00544-t001:** Particle properties of the LNPs with different amounts of PEG–lipid in Figure 3a.

Sample	Size (nm)	PdI (Polydispersity Index)	Zeta-Potential (mV)	siRNA Recovery	siRNA Encapsulation
0.5% PEG	185	0.124	−5.7	68%	99%
1.0% PEG	121	0.121	−4.5	65%	100%
1.5% PEG	112	0.135	−4.6	64%	100%
2.0% PEG	101	0.184	−3.5	69%	100%
2.5% PEG	119	0.120	−3.2	79%	100%
3.0% PEG	91	0.121	−2.5	68%	100%

**Table 2 pharmaceutics-13-00544-t002:** Particle properties of the siRNA-encapsulating LNPs with different ssPalm (as shown in Figure 5).

Lipid	Size (nm)	PdI	Zeta-Potential (mV)	siRNA Recovery	siRNA Encapsulation
ssPalmO-Phe	96	0.079	−6.0	46%	100%
ssPalmO-Ben	98	0.069	−6.5	51%	99%
ssPalmO-P4C2	98	0.099	−7.0	21%	100%
ssPalmE-P4C2	104	0.057	−6.3	60%	100%

**Table 3 pharmaceutics-13-00544-t003:** Particle properties of the LNPs with different antisense oligonucleotide (ASO) in Figure 8.

Oligonucleotides	Size (nm)	PdI	Zeta-Potential (mV)	ASO Recovery ^1^
ASOap	45	0.156	−2.6	24.1%
ASOgr	53	0.185	+12	71.4%

^1^ Encapsulation efficiency of the ASO was not calculated accurately since the OliGreen^TM^ assay of LNP–ASO in the absence of surfactant cannot produce reliable fluorescence.

## Data Availability

Not applicable.

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
