# Peer review of "Delivery of Oligonucleotides Using a Self-Degradable Lipid-Like Material"

_pharmaceutics, 2021, doi:10.3390/pharmaceutics13040544_

Round 1

Reviewer 1 Report

In this manuscript, the authors explored the optimal composition of an LNP containing a 19 self-degradable material (ssPalmO-Phe) for the delivery of oligonucleotides. The ssPalmO-Phe-LNP system was originally developed for mRNA delivery, and the authors optimized this system for efficient delivery of ASO and siRNA. Importantly, the authors found that the composition of the LNPs needed to be modified to achieve optimal delivery efficiency for different types of nucleic acids. Overall, this study provided a systematic understanding of how the formulation of LNP influenced their delivery efficiency of different nucleic acids, which is fundamentally important and could benefit the development of the more efficient LNP systems for the delivery of various types of nucleic acids. The reviewer would therefore recommend publication of the manuscript in Pharmaceutics after the following comments are addressed.

1, The authors optimized an mRNA-LNP system for siRNA and ASO delivery. What is the principle or logic for the optimization of LNP for different kinds of nucleic acids?

2, In Table 3, The ASOap and ASOgr are two negative-charged molecules with similar molecular weight, why the LNP contains these two antisense showed different surface charges? Please clarify.

3, In Figure 7, from (a) to (c), the conc. of protein increased, however, the amounts of MC3 and ssPalmO-Phe also changed, how could the authors draw any conclusions since two parameters changed at the same time. Please explain.

4, On Page 3, line 125, there is an error for the unit (7.5 ?g/mL), and this kind of errors are everywhere in the manuscript. Please clarify.

Author Response

Thank you very much for careful reading of our paper. Our replies to your inquiries are summarized in the attachment.

Reviewer 2 Report

Tanaka et al performed a study on the optimization of degradable lipid compositions of ssPalm based systems. It certainly would have been interesting to benchmark this against the non-degradable system, but I am not indicating that is absolutely essential for the publication of this manuscript. The authors need to correct the symbols throughout the manuscript that have possibly gotten messed up during the formatting of the manuscript. Also, the clarity of some of the figures such as figures 8 and 9 needs to be improved. I can approve of the publication of this manuscript once these issues have been fixed.

Author Response

(The authors gave the same response as above.)

Reviewer 3 Report

I have read carefully the work entitled “Delivery of oligonucleotides using self-degradable lipid like material”.

The manuscript describes in detail the optimization procedure of liposomes to deliver siRNA.

In my opinion, the paper contains interesting information and clear results but I have some comments or doubts to answer:

Major issues:

-In my opinion the characterization of the liposomes used should appear before the in vivo results in all sections.

-The reference to the figures in Material and methods is confusing as it seems to show the experimental design and they show the results instead.

-Is TritonX100 used during the quantification of siRNA in all cases? This is not clearly explained in the methodology.

-Did the authors quantify the lipid amount? Please explain the calculation of encapsulation efficiency.

-The authors say that the intravenous injected volume was adjusted to 10 ml/kg. This means 210 µl for a 21 gr mouse. In my opinion this volume is excessive. Please comment.

-How many mice were used in each experiment?

-In general the results section contains lot of information that should be mentioned in methods.

-I found confusing to follow the total amount of PEG in each formulation.

-Why the 1% peg formulation was not considered appropriate? The difference in size is compensated by the FVII serum concentration.

-What is the reason of the change in the dose from 0.02 mg/kg to 0.01 mg/kg in figure 3?

-Did the authors perform a toxicity evaluation during the evaluation of ED50 value?

-I do not understand how the hemolytic activity relates with the endosomal escape efficiency. In my opinion this issue was not deducible from the data.

-In the determination of IC50 the administered amounts are sometimes above the 0.37 mg/kg that are related with severe toxicity as mentioned in line 411. Please comment on that.

-Although not required for the paper, a pharmacokinetic evaluation of the formulations would improve the quality of the work.

Minor issues:

-What is the meaning of ODN (line 25)?

-The symbols µ and ζ are not properly introduced in the text as a weird symbol appears instead.

-Please detail the ultracentrifugation conditions

-What is the meaning of TNS assay? (line 324)

-The discussion repeats the results too much.

Author Response

(The authors gave the same response as above.)
